# Sharp bounds on aggregate expert error

**Aryeh Kontorovich**                                                    KARYEH@CS.BGU.AC.IL
*Computer Science Department, Ben-Gurion University of the Negev Beer Sheva, Israel*

**Ariel Avital**                                                         AVITALQ@POST.BGU.AC.IL
*Computer Science Department, Ben-Gurion University of the Negev Beer Sheva, Israel*

**Editors:** Gautam Kamath and Po-Ling Loh

## Abstract

We revisit the classic problem of aggregating binary advice from conditionally independent experts, also known as the Naive Bayes setting. Our quantity of interest is the error probability of the optimal decision rule. In the case of symmetric errors (sensitivity = specificity), reasonably tight bounds on the optimal error probability are known. In the general asymmetric case, we are not aware of any nontrivial estimates on this quantity. Our contribution consists of sharp upper and lower bounds on the optimal error probability in the general case, which recover and sharpen the best known results in the symmetric special case. Additionally, our bounds are apparently the first to take the bias into account. Since this turns out to be closely connected to bounding the total variation distance between two product distributions, our results also have bearing on this important and challenging problem.

**Keywords:** experts, hypothesis testing, Neyman-Pearson lemma , naive Bayes

## 1. Introduction

Consider the following decision-theoretic setting. A parameter $\theta \in (0,1)$ is fixed and a random bit $Y \in \{0,1\}$ is drawn according to Bernoulli with bias $\theta$: that is, $\theta = \mathbb{P}(Y = 1) = 1 - \mathbb{P}(Y = 0)$. Conditional on $Y$, the $\{0,1\}$-valued variables $X_1, X_2, \ldots, X_n$ are drawn independently according to

$$\mathbb{P}(X_i = 1 | Y = 1) \quad = \quad \psi_i, \tag{1}$$
$$\mathbb{P}(X_i = 0 | Y = 0) \quad = \quad \eta_i \tag{2}$$

for some collection of parameters $\psi, \eta \in (0,1)^n$. The $\psi_i$ and $\eta_i$ are classically known as *sensitivity* and *specificity*, respectively. An agent who knows the values of $\theta, \psi, \eta$ gets to observe $X = (X_1, \ldots, X_n)$ and wishes to infer the most likely $Y$ conditional on $X$. A decision rule $f : \{0,1\}^n \to \{0,1\}$ that minimizes the *error probability* $\mathbb{P}(f(X) \neq Y)$ may be found in Parisi et al. (2014, Eqs. (11), (12)):[1]

$$f^{\mathsf{OPT}} : X \mapsto \mathrm{sign}\left( \log \frac{\theta}{1-\theta} + \sum_{i=1}^{n} (2X_i - 1) \log \alpha_i + \log \beta_i \right), \tag{3}$$

where

$$\alpha_i = \frac{\psi_i \eta_i}{(1 - \psi_i)(1 - \eta_i)}, \quad \beta_i = \frac{\psi_i(1 - \psi_i)}{\eta_i(1 - \eta_i)},$$

---

1. The bound therein was stated for the case $\theta = 1/2$ — i.e., without the $\log \frac{\theta}{1-\theta}$ term. We rederive the full expression for completeness in Section 4.1.

and $\mathrm{sign}$, along with the rest of our notation, is defined below.

The main quantity of interest in this note is the optimal error probability $\mathbb{P}(f^{\mathsf{OPT}}(X) \neq Y)$. We obtain improved, sharp bounds on this quantity in the symmetric ($\psi = \eta$) and asymmetric cases. It will turn out that estimating $\mathbb{P}(f^{\mathsf{OPT}}(X) \neq Y)$ is closely related to computing the total variation distance between two product distributions — and thus our results also have bearing on this important and computationally challenging problem.

**Motivation.** The Neyman-Pearson Lemma (see (8)) lies at the heart of decision theory and hypothesis testing, as it provides an optimal risk-minimizing strategy. Our results continue a line of work that analyzes the *performance* of this optimal strategy. Plans for future work include finite-sample guarantees based on the spectral estimates of Parisi et al. (2014).

**Definitions.** The *balanced accuracy* is defined as $\pi_i = (\psi_i + \eta_i)/2$. We will consistently use the notation $\bar{\varphi} := 1 - \varphi$ for all expressions $\varphi$; thus, in particular $\bar{p} = 1 - p$ and $\overline{1-p} = p$. For $p \in (0,1)$, we write $\mathrm{Ber}(p)$ to denote the Bernoulli measure on $\{0,1\}$; that is, $\mathrm{Ber}(p)(0) = \bar{p}$, $\mathrm{Ber}(p)(1) = p$. For $n \in \mathbb{N}$ and $p = (p_1, \ldots, p_n) \in (0,1)^n$, $\mathrm{Ber}(p)$ denotes the product of $n$ Bernoulli distributions with parameters $p_i$:

$$\mathrm{Ber}(p) \quad := \quad \mathrm{Ber}(p_1) \otimes \mathrm{Ber}(p_2) \otimes \ldots \otimes \mathrm{Ber}(p_n).$$

Thus, $\mathrm{Ber}(p)$ is a probability measure on $\{0,1\}^n$, with

$$\mathrm{Ber}(p)(x) = \prod_{i=1}^{n} p_i^{x_i}(1-p_i)^{1-x_i} = \prod_{i=1}^{n} p_i^{x_i} \bar{p}_i^{\bar{x}_i}, \qquad x \in \{0,1\}^n,$$

We write $[n] := \{1, \ldots, n\}$ and use standard vector norm notation $\|w\|_p^p = \sum_{i \in [n]} |w_i|^p$ for $w \in \mathbb{R}^n$. For probability measures $P, Q$ on a finite set $\Omega$, their total variation distance is

$$\|P - Q\|_{\mathsf{TV}} := \frac{1}{2} \|P - Q\|_1 = \frac{1}{2} \sum_{x \in \Omega} |P(x) - Q(x)|.$$

We will make use of Scheffé's identity (Tsybakov, 2009, Lemma 2.1):

$$\|P \wedge Q\|_1 \;=\; 1 - \|P - Q\|_{\mathsf{TV}}, \tag{4}$$

where $u \wedge v = \min\{u, v\}$ and $\sqrt{PQ}$, $P \wedge Q$ are shorthands for the measures on $\Omega$ given by $\sqrt{P(x)Q(x)}$ and $P(x) \wedge Q(x)$ respectively. For $t \in \mathbb{R}$, $\mathrm{sign}(t) := \mathbf{1}[t \geq 0] - \mathbf{1}[t < 0]$.

**Remark 1** *The issue of optimally breaking ties in (3) is somewhat delicate and is exhaustively addressed in Kontorovich and Pinelis (2019, Eq. (2.7)). Fortunately, although there may be several optimal decision rules, they all share the same minimum probability of error, which depends continuously on the parameters $\theta$, $\psi$, $\eta$. Thus, we can always infinitesimally perturb these so as to avoid ties, and assume no ties henceforth.*

## 2. Background and related work

We refer the reader to Parisi et al. (2014) and Berend and Kontorovich (2015) for a detailed background and literature review of this problem. Parisi et al. and Zhang et al. (2014) proposed a

spectral method for inferring the accuracy of the experts from unsupervised data only. Follow-up works include Jaffe et al. (2015, 2016); Shaham et al. (2016); Tenzer et al. (2022).

For the case of *symmetric* experts with $\psi = \eta =: p$ and $\theta = 1/2$, the optimal rule $f^{\mathsf{OPT}}$ given in (3) reduces to $f^{\mathsf{OPT}}(X) = \mathrm{sign}\left(\sum_{i=1}^n w_i X_i\right)$, where $w_i := \log \frac{p_i}{\bar{p}_i}$. Berend and Kontorovich (2015) showed that

$$\mathbb{P}(f^{\mathsf{OPT}} \neq Y) \;\; = \;\; \frac{1}{2}\,\|\mathrm{Ber}(p) \wedge \mathrm{Ber}(\bar{p})\|_1 \tag{5}$$

and, putting $\Phi = \sum_{i=1}^n \left(p_i - \frac{1}{2}\right) w_i$, Theorem 1 therein states that

$$\frac{3}{4[1 + \exp(2\Phi + 4\sqrt{\Phi})]} \;\; \leq \;\; \mathbb{P}(f^{\mathsf{OPT}} \neq Y) \;\; \leq \;\; \exp(-\Phi/2). \tag{6}$$

Follow-up works include Gao et al. (2016) and Manino et al. (2019). In particular, Manino et al., (Theorem 1, Theorem 3) showed[2] that

$$0.36 \cdot 2^n \sqrt{\prod_{i=1}^n p_i \bar{p}_i} \cdot \exp\left(-\frac{1}{2}\sqrt{\sum_{i=1}^n w_i^2}\right) \leq \mathbb{P}(f^{\mathsf{OPT}} \neq Y) \leq \frac{1}{2} \cdot 2^n \sqrt{\prod_{i=1}^n p_i \bar{p}_i} \tag{7}$$

and further demonstrated that (7) sharpens both estimates in (6).

## 3. Main results

We begin with an analog of (5) generalized in two ways: the experts are neither assumed to be symmetric (sensitivity = specificity) nor unbiased ($\theta = 1/2$) — and in particular, $\theta$ explicitly figures in the expressions.

**Theorem 1** *For conditionally independent experts as in (1,2) with sensitivities $\psi$ and specificities $\eta$, the decision rule $f^{\mathsf{OPT}}$ in (3) satisfies*

$$\mathbb{P}(f^{\mathsf{OPT}} \neq Y) \;\; = \;\; \left\|\theta P \wedge \bar{\theta} Q\right\|_1,$$

*where $P = \mathrm{Ber}(\psi)$ and $Q = \mathrm{Ber}(\bar{\eta})$.*

Next, we provide an upper bound on $f^{\mathsf{OPT}}$ in terms of the balanced accuracy $\pi_i = (\psi_i + \eta_i)/2$:

**Theorem 2** *Under the conditions of Theorem 1,*

$$\mathbb{P}(f^{\mathsf{OPT}} \neq Y) \;\; \leq \;\; \sqrt{\theta\bar{\theta} \prod_{i=1}^n (\psi_i + \eta_i)(2 - \psi_i - \eta_i)} = 2^n \sqrt{\theta\bar{\theta} \prod_{i=1}^n \pi_i \bar{\pi}_i}.$$

The above sharpens the upper bound for the symmetric case $\psi = \eta = p$ in (7) for asymmetric bias (while recovering it for $\theta = 1/2$). An additional interesting limiting case is where $\psi_i = \bar{\eta}_i$ for all $i \in [n]$. In this case, the experts contribute nothing, and our upper bound evaluates to $\sqrt{\theta\bar{\theta}}$. While this gives the exact error for $\theta = \frac{1}{2}$, it is loose for $\theta$ close to 0 or 1. At the other extreme, if at least one of the $\pi_i \in \{0, 1\}$ (which can only happen if the corresponding $\psi_i = \eta_i = \pi_i$), the bound evaluates to 0, as it should.

Our next result is a lower bound on the error probability:

---

2. Manino et al. acknowledge the priority of Gao et al. (2016) for both bounds in (7), improving their constant in the lower bound from 0.25 to 0.36.

**Theorem 3** *Under the conditions of Theorem 1,*

$$\mathbb{P}(f^{\mathsf{OPT}} \neq Y) \;\geq\; \min\{\theta, \bar{\theta}\} \cdot 2^n \sqrt{\prod_{i=1}^{n} \pi_i \bar{\pi}_i} \cdot \exp\left(-\frac{1}{2}\sum_{i=1}^{n} |\gamma_i|\right),$$

*where $\gamma_i = \log(\pi_i/\bar{\pi}_i)$.*

Note that the factor $\sqrt{\theta\bar{\theta}}$ in the upper bound cannot be sharpened to match the factor $\min\{\theta, \bar{\theta}\}$ in the lower bound. This is demonstrated by taking $n = 1$ and $\psi_1 = \eta_1 = \theta \neq 1/2$. In this case, $\mathbb{P}(f^{\mathsf{OPT}} \neq Y) = \theta$, while such a putative upper bound would evaluate to $2\theta\sqrt{\theta\bar{\theta}} < \theta$.

In the symmetric case, the lower bound can be sharpened:

**Theorem 4** *Under the conditions of Theorem 1, where also $\psi = \eta = p$,*

$$\mathbb{P}(f^{\mathsf{OPT}} \neq Y) \;\geq\; \min\left\{\theta, \bar{\theta}\right\} \cdot 2^n \sqrt{\prod_{i=1}^{n} p_i \bar{p}_i} \cdot \exp\left(-\frac{1}{2}\sqrt{\sum_{i=1}^{n} w_i^2}\right),$$

*where $w_i = \log(p_i/\bar{p}_i)$.*

Since $\gamma = w$ in the symmetric case and $\|w\|_2 \leq \|w\|_1 \leq \sqrt{n}\,\|w\|_2$, the bound in Theorem 4 is indeed significantly sharper than that in Theorem 3. To illustrate sharpness, consider the case where $p_i = 1/2$, for all $i \in [n]$. In this case, the bound evaluates to $\min\{\theta, \bar{\theta}\}$, which is the exact value of $\mathbb{P}(f^{\mathsf{OPT}} \neq Y)$.

**Remark 5** *This bound is sufficiently sharp to yield the bound $\|\mathrm{Ber}(p) - \mathrm{Ber}(\bar{p})\|_{\mathsf{TV}} \leq \|p - \bar{p}\|_2$ with the optimal constant* 1, *(Kontorovich, 2024, Theorem 3).*

**Tightness and counterexamples.** In this subsection, we take $\theta = 1/2$. Theorem 2 is loose in the regime $n = 1$, $p = \varepsilon$ for small $\varepsilon$; here, $\mathbb{P}(f^{\mathsf{OPT}} \neq Y) \sim \varepsilon$, while the bound is $\sim \sqrt{\varepsilon}$. One might be tempted to improve the $\|\gamma\|_1$ appearing in the bound of Theorem 3 to the sharper value $\|\gamma\|_2$. Unfortunately, that sharper bound does not hold. Indeed, take $n = 2$, $\psi = (1, 0)$, and $\eta = (1 - \varepsilon, \varepsilon)$ for $\varepsilon \in (0, 1)$. Then $\pi = (1 - \varepsilon/2, \varepsilon/2)$ and $\gamma = (\log(2/\varepsilon - 1), \log(\varepsilon/(2-\varepsilon))$. It is straightforward to verify that $\|\mathrm{Ber}(\psi) \wedge \mathrm{Ber}(\bar{\eta})\|_1 = \varepsilon^2$ and that

$$\sqrt{\prod_{i=1}^{n} \pi_i \bar{\pi}_i}\, \mathrm{e}^{-\frac{1}{2}\|\gamma\|_2} \;=\; \frac{\varepsilon}{2}\left(1 - \frac{\varepsilon}{2}\right)\exp\left(-\frac{\log(2/\varepsilon - 1)}{\sqrt{2}}\right)$$

$$\;=\; \frac{\varepsilon}{2}\left(1 - \frac{\varepsilon}{2}\right)\left(\frac{\varepsilon}{2 - \varepsilon}\right)^{1/\sqrt{2}} =: f(\varepsilon).$$

Since $f(\varepsilon)/\varepsilon^2 \to \infty$ as $\varepsilon \downarrow 0$, we conclude that the conjectural bound fails to hold.

We can also exhibit a regime in which Theorem 4 is not tight. Take $n = 2$ and $p = (\varepsilon, \varepsilon)$ for $0 < \varepsilon < 1/2$. Then $\|\mathrm{Ber}(p) \wedge \mathrm{Ber}(\bar{p})\|_1 = 2\varepsilon$, $w = (\log(\varepsilon/(1 - \varepsilon), \log((1 - \varepsilon)/\varepsilon)$, and

$$\sqrt{\prod_{i=1}^{n} p_i \bar{p}_i}\, \mathrm{e}^{-\frac{1}{2}\|w\|_2} \;=\; \varepsilon(1 - \varepsilon)\exp\left(-\frac{\log(1/\varepsilon - 1)}{\sqrt{2}}\right)$$

$$\;=\; \varepsilon(1 - \varepsilon)\left(\frac{\varepsilon}{1 - \varepsilon}\right)^{1/\sqrt{2}} =: g(\varepsilon).$$

Since $g(\varepsilon)/\varepsilon \to 0$ as $\varepsilon \downarrow 0$, the bound is quite loose in this regime.

**Algorithmic aspects.** Bhattacharyya et al. (2023) showed that for general $p, q \in [0,1]^n$, it is hard to compute $\|\mathrm{Ber}(p) - \mathrm{Ber}(q)\|_{\mathsf{TV}}$ exactly. Feng et al. (2023) gave an efficient randomized algorithm for obtaining a $1 \pm \varepsilon$ multiplicative approximation with confidence $\delta$, in time $O(\frac{n^2}{\varepsilon^2} \log \frac{1}{\delta})$; this was later derandomized by Feng et al. (2024). Since our results approximate $\|P \wedge Q\|_1 = 1 - \|P - Q\|_{\mathsf{TV}}$, they are not directly comparable. Note also that our bounds in Theorems 2, 3 are stated in terms of $p - q$ in simple, analytically tractable closed formulas. Still, as discussed above, certain gaps between the upper and lower bounds persist, and one is led to wonder to what extent these are due to computational hardness obstructions.

## 4. Proofs

We maintain our convention $\bar{\varphi} = 1 - \varphi$ for all expressions $\varphi$.

### 4.1. Proof of Theorem 1

The Neyman-Pearson lemma (Cover and Thomas, 2006, Theorem 11.7.1) implies that $f^{\mathsf{OPT}}$ must satisfy

$$\mathbb{P}(f^{\mathsf{OPT}}(X) = Y | X = x) \;\geq\; \mathbb{P}(f^{\mathsf{OPT}}(X) \neq Y | X = x), \qquad x \in \{0,1\}^n. \tag{8}$$

By the Bayes formula, an equivalent condition is that $f^{\mathsf{OPT}}(x) = 1$ if and only if

$$\theta \prod_{i \in A} \psi_i \prod_{i \in B} \bar{\psi}_i \;\geq\; \bar{\theta} \prod_{i \in A} \bar{\eta}_i \prod_{i \in B} \eta_i, \tag{9}$$

where $A, B \subseteq [n]$ are given by $A = \{i \in [n] : x_i = 1\}$ and $B = \{i \in [n] : x_i = 0\}$. Taking logarithms, (9) is equivalent to

$$\log \frac{\theta}{\bar{\theta}} + \sum_{i=1}^{n} x_i \log \frac{\psi_i}{\bar{\eta}_i} + \sum_{i=1}^{n} \bar{x}_i \log \frac{\bar{\psi}_i}{\eta_i} \geq 0; \tag{10}$$

this is easily seen to be equivalent to (3). Now,

$$\mathbb{P}(f^{\mathsf{OPT}}(X) \neq Y) = \theta \mathbb{P}(f^{\mathsf{OPT}}(X) \neq Y | Y = 1) + \bar{\theta} \mathbb{P}(f^{\mathsf{OPT}}(X) \neq Y | Y = 0).$$

Conditional on $Y = 1$, define the random variables $Z_i = \mathbf{1}[X_i = Y]$ and note that the $(Z_1, \ldots, Z_n)$ are jointly distributed according to $P = \mathrm{Ber}(\psi)$. Putting $Q = \mathrm{Ber}(\bar{\eta})$, (9) implies that when $Y = 1$, $f^{\mathsf{OPT}}$ makes a mistake on $x \in \{0,1\}^n$ precisely[3] when $\theta P(x) < \bar{\theta} Q(x)$, whence

$$\mathbb{P}(f^{\mathsf{OPT}}(X) \neq Y | Y = 1) \;=\; \sum_{x \in \{0,1\}^n} P(x) \mathbf{1}[\theta P(x) < \bar{\theta} Q(x)]. \tag{11}$$

A similar analysis shows that

$$\mathbb{P}(f^{\mathsf{OPT}}(X) \neq Y | Y = 0) = \sum_{x \in \{0,1\}^n} Q(x) \mathbf{1}[\theta P(x) \geq \bar{\theta} Q(x)]. \tag{12}$$

Since $u \mathbf{1}[u < v] + v \mathbf{1}[v \leq u] = u \wedge v$, we have

$$\mathbb{P}(f^{\mathsf{OPT}}(X) \neq Y) \;=\; \sum_{x \in \{0,1\}^n} \theta P(x) \wedge \bar{\theta} Q(x) = \left\| \theta P \wedge \bar{\theta} Q \right\|_1.$$

which finishes the proof. ∎

---

3. As per Remark 1, there is no loss of generality in assuming no ties.

## 4.2. Proof of Theorem 2

The following result may be of independent interest. Only the upper bound is used in this paper.

**Lemma 1** *For $p, q \in (0, 1)^n$, let $P, Q$ be two probability measures on $\{0, 1\}^n$ given by $P = \mathrm{Ber}(p)$ and $Q = \mathrm{Ber}(q)$. Then*

$$\sqrt{\prod_{i=1}^{n} \frac{1 - (p_i - q_i)^2}{2}} \leq \sum_{x \in \{0,1\}^n} \sqrt{P(x)Q(x)} \leq \sqrt{\prod_{i=1}^{n} [1 - (p_i - q_i)^2]}.$$

**Proof** We prove both inequalities by induction on $n$, starting with the second. The base case, $n = 1$, amounts to showing that

$$\sqrt{st} + \sqrt{(1-s)(1-t)} \leq \sqrt{1 - (s-t)^2}, \qquad s, t \in (0, 1). \tag{13}$$

Squaring both sides, (13) is equivalent to

$$1 - s - t + 2st + 2\sqrt{st(1-s)(1-t)} \leq 1 - (s-t)^2,$$

which, after canceling like terms, simplifies to

$$2\sqrt{st(1-s)(1-t)} \leq s - s^2 + t - t^2. \tag{14}$$

Denoting the right-hand-side of (14) by $R$ and the left-hand-side by $L$, we compute

$$R^2 - L^2 = (s - s^2 - t + t^2)^2 \geq 0,$$

which proves (13). Now we assume that the claim holds for some $n = k$ and consider the case $n = k + 1$:

$$\sum_{x \in \{0,1\}^k, y \in \{0,1\}} \sqrt{P(x, y)Q(x, y)} = \sum_{x \in \{0,1\}^k} \sqrt{P(x, 0)Q(x, 0)} + \sum_{x \in \{0,1\}^k} \sqrt{P(x, 1)Q(x, 1)}$$

$$= \sum_{x \in \{0,1\}^k} \sqrt{P(x)Q(x)(1 - p_{k+1})(1 - q_{k+1})}$$

$$+ \sum_{x \in \{0,1\}^k} \sqrt{P(x)Q(x)p_{k+1}q_{k+1}}.$$

Now apply the inductive hypothesis to each term:

$$\sum_{x \in \{0,1\}^k} \sqrt{P(x)Q(x)p_{k+1}q_{k+1}} = \sqrt{p_{k+1}q_{k+1}} \sum_{x \in \{0,1\}^k} \sqrt{P(x)Q(x)}$$

$$\leq \sqrt{p_{k+1}q_{k+1} \prod_{i=1}^{k} [1 - (p_i - q_i)^2]}$$

(the analogous bound holds for the other term). Putting $s = p_{k+1}$, $t = q_{k+1}$, and $K = \prod_{i=1}^{k}[1 - (p_i - q_i)^2]$, we obtain

$$\sum_{x \in \{0,1\}^{k+1}} \sqrt{P(x)Q(x)} \leq \sqrt{stK} + \sqrt{(1-s)(1-t)K}$$

$$\leq \sqrt{(1+s-t)(1+t-s)K} = \sqrt{\prod_{i=1}^{k+1}[1 - (p_i - q_i)^2]},$$

where (13) was invoked in the second inequality. This proves the upper bound on $\sum \sqrt{P(x)Q(x)}$.

The lower bound proceeds in an entirely analogous fashion, only with

$$\sqrt{st} + \sqrt{(1-s)(1-t)} \geq \sqrt{\frac{1 - (s-t)^2}{2}}, \qquad s, t \in (0,1) \tag{15}$$

as the base case instead of (13). To prove (15), recall that $\sqrt{u+v} \leq \sqrt{u} + \sqrt{v}$ for $u, v \geq 0$ to obtain the stronger inequality

$$\sqrt{st + (1-s)(1-t)} \geq \sqrt{\frac{1 - (s-t)^2}{2}}.$$

Squaring both sides and collecting terms yields the equivalent (and obviously true) $(1 - s - t)^2 \geq 0$. From here, the induction proceeds exactly as in the upper bound: we put $s = p_{k+1}$, $t = q_{k+1}$, and $K = \prod_{i=1}^{k}[1 - (p_i - q_i)^2]$, and repeat the steps therein with the inequality appropriately reversed. ∎

**Proof** [of Theorem 2] By Theorem 1, $\mathbb{P}(f^{\mathrm{OPT}} \neq Y) = \left\|\theta P \wedge \bar{\theta}Q\right\|_1$. Now since $a \wedge b \leq \sqrt{ab}$ for $a, b \geq 0$, we have

$$\sum_{x \in \{0,1\}^n} P'(x) \wedge Q'(x) \leq \sum_{x \in \{0,1\}^n} \sqrt{P'(x)Q'(x)} \tag{16}$$

for all positive measures $P', Q'$ on $\{0,1\}^n$. Setting $P' = \theta P$ and $Q' = \bar{\theta}Q$, we now have that

$$\sum_{x \in \{0,1\}^n} \theta P(x) \wedge \bar{\theta}Q(x) \leq \sqrt{\theta\bar{\theta}} \sum_{x \in \{0,1\}^n} \sqrt{P(x)Q(x)}. \tag{17}$$

Applying the upper bound in Lemma 1 with $p_i = \psi_i$ and $q_i = 1 - \eta_i$ and noting that $1 - (\psi_i - \bar{\eta}_i)^2 = 4\pi_i\bar{\pi}_i$ completes the proof. ∎

### 4.3. Proof of Theorem 3

As $\mathbb{P}(f^{\mathrm{OPT}} \neq Y) = \left\|\theta P \wedge \bar{\theta}Q\right\|_1$, we start by writing

$$\left\|\theta P \wedge \bar{\theta}Q\right\|_1 \geq \min\left\{\theta, \bar{\theta}\right\} \|P \wedge Q\|_1 = \min\left\{\theta, \bar{\theta}\right\} \left\|\mathrm{Ber}(\psi) \wedge \mathrm{Ber}(\bar{\eta})\right\|_1,$$

where we used the pointwise inequality

$$\min\left\{\lambda u, \bar{\lambda} v\right\} \geq \min\left\{\lambda, \bar{\lambda}\right\} \min\left\{u, v\right\}.$$

Next, we invoke Lemma 2 with $P = \mathrm{Ber}(\psi)$ and $Q = \mathrm{Ber}(\bar{\eta})$ to obtain

$$
\begin{aligned}
\|\mathrm{Ber}(\psi) \wedge \mathrm{Ber}(\bar{\eta})\|_1 &\geq \prod_{i=1}^{n} \|\mathrm{Ber}(\psi_i) \wedge \mathrm{Ber}(\bar{\eta}_i)\|_1 \\
&= \prod_{i=1}^{n} \left[\psi_i \wedge \bar{\eta}_i + \bar{\psi}_i \wedge \eta_i\right] \\
&\geq \prod_{i=1}^{n} 2(\pi_i \wedge \bar{\pi}_i),
\end{aligned}
$$

where the last inequality is due to (20). By (19),

$$\pi_i \wedge \bar{\pi}_i = \sqrt{\pi_i \bar{\pi}_i} \exp\left(-\frac{1}{2}\left|\log\frac{\pi_i}{\bar{\pi}_i}\right|\right),$$

whence

$$\prod_{i=1}^{n} 2(\pi_i \wedge \bar{\pi}_i) = 2^n \sqrt{\prod_{i=1}^{n} \pi_i \bar{\pi}_i} \cdot \exp\left(-\frac{1}{2}\sum_{i=1}^{n}\left|\log\frac{\pi_i}{\bar{\pi}_i}\right|\right).$$

This proves the claim. ∎

### 4.4. Proof of Theorem 4

Repeating the argument from Theorem 3, we have

$$\mathbb{P}(f^{\mathsf{OPT}} \neq Y) \geq \min\left\{\theta, \bar{\theta}\right\} \|\mathrm{Ber}(\psi) \wedge \mathrm{Ber}(\bar{\eta})\|_1. \tag{18}$$

Setting $\psi_i = \eta_i = p_i$ we invoke (19) to obtain

$$
\begin{aligned}
\|\mathrm{Ber}(p) \wedge \mathrm{Ber}(\bar{p})\|_1 &= \sum_{x \in \{0,1\}^n} P(x) \wedge Q(x) \\
&= \sum_{x \in \{0,1\}^n} \sqrt{P(x)Q(x)} \exp\left(-\frac{1}{2}\left|\log\frac{P(x)}{Q(x)}\right|\right) \\
&= \sqrt{\prod_{i=1}^{n} p_i \bar{p}_i} \sum_{x \in \{0,1\}^n} \exp\left(-\frac{1}{2}\left|\log\frac{P(x)}{Q(x)}\right|\right) \\
&= 2^n \sqrt{\prod_{i=1}^{n} p_i \bar{p}_i} \mathop{\mathbb{E}}_{Z \sim \mathrm{Uniform}\{0,1\}^n} \exp\left(-\frac{1}{2}\left|\log\frac{P(Z)}{Q(Z)}\right|\right) \\
&\geq 2^n \sqrt{\prod_{i=1}^{n} p_i \bar{p}_i} \exp\left(-\frac{1}{2}\mathop{\mathbb{E}}_{Z}\left|\log\frac{P(Z)}{Q(Z)}\right|\right),
\end{aligned}
$$

where Jensen's inequality was used in the last step. Since $P, Q$ are product measures, we have $P(Z) = \prod_{i=1}^{n} P_i(Z_i)$ and $Q(Z) = \prod_{i=1}^{n} Q_i(Z_i) = \prod_{i=1}^{n} \bar{P}_i(Z_i)$, whence

$$
\begin{aligned}
\mathbb{E}\left|\log \frac{P(Z)}{Q(Z)}\right| &\leq \sqrt{\mathbb{E}\left[\left(\log \frac{P(Z)}{Q(Z)}\right)^2\right]} \\
&= \sqrt{\mathbb{E}\left[\left(\sum_{i=1}^{n} \log \frac{P_i(Z_i)}{\bar{P}_i(Z_i)}\right)^2\right]} \\
&= \sqrt{\mathbb{E}\left[\sum_{i,j \in [n]} \log \frac{P_i(Z_i)}{\bar{P}_i(Z_i)} \log \frac{P_j(Z_j)}{\bar{P}_j(Z_j)}\right]}.
\end{aligned}
$$

Since $\mathbb{E} \log \frac{P_i(Z_i)}{\bar{P}_i(Z_i)} = 0$ and the $Z_i$ are independent, only the diagonal terms survive:

$$
\begin{aligned}
\mathbb{E}\left[\sum_{i,j \in [n]} \log \frac{P_i(Z_i)}{\bar{P}_i(Z_i)} \log \frac{P_j(Z_j)}{\bar{P}_j(Z_j)}\right] &= \sum_{i=1}^{n} \mathbb{E}\left|\log \frac{P_i(Z_i)}{\bar{P}_i(Z_i)}\right|^2 \\
&= \sum_{i=1}^{n} \left(\log \frac{p_i}{\bar{p}_i}\right)^2.
\end{aligned}
$$

The proof is complete. ∎

### 4.5. Auxiliary Lemmata

The following identity and inequality are elementary:

$$
u \wedge v = \sqrt{uv} \exp\left(-\frac{1}{2}\left|\log \frac{u}{v}\right|\right), \qquad u, v > 0, \tag{19}
$$

$$
s \wedge \bar{t} + t \wedge \bar{s} \geq 2(u \wedge \bar{u}), \qquad s, t \in [0,1], u = (s+t)/2. \tag{20}
$$

**Lemma 2** *For all probability measures $P, Q, P', Q'$, we have*

$$
\left\|P \otimes Q \wedge P' \otimes Q'\right\|_1 \geq \left\|P \wedge P'\right\|_1 \cdot \left\|Q \wedge Q'\right\|_1.
$$

**Proof** It is a classic fact (see, e.g., Kontorovich (2012, Lemma 2.2)) that

$$
\left\|P \otimes Q - P' \otimes Q'\right\|_{\mathsf{TV}} \leq \left\|P - P'\right\|_{\mathsf{TV}} + \left\|Q - Q'\right\|_{\mathsf{TV}} - \left\|P - P'\right\|_{\mathsf{TV}} \cdot \left\|Q - Q'\right\|_{\mathsf{TV}}.
$$

The claim follows by Scheffé's identity (4). ∎

## Acknowledgments

We thank Daniel Berend, Lior Daniel, Ariel Jaffe, Douglas Hubbard, Sudeep Kamath, Mark Kozlenko, Kuldeep Meel, Dimitrios Myrisiotis, Boaz Nadler, and Rotem Zur for enlightening discussions.

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
