# OpenReview forum: "Sharp bounds on aggregate expert error"
_algorithmiclearningtheory.org/ALT/2025/Conference — ALT 2025_

### Official Review · Reviewer_7DKp · 2024-11-05
**Review for the Paper #10**

**Rating:** 6
**Confidence:** 4

**Review:**

**Summary:** This paper examines the learnability—specifically, the lower and upper bounds on the optimal decision rule—for aggregating binary predictions from conditionally independent experts. The authors' key contribution lies in establishing sharp upper and lower bounds on the total variation distance between two product Bernoulli distributions with arbitrary means.

**Strength:**

1. **Theoretical Contribution with New Bounds**: This paper provides new upper and lower bounds on error probabilities in the asymmetric case, extending previous work on aggregate expert error. These bounds contribute to a deeper understanding of error probability in cases where sensitivity and specificity differ, addressing a previously less-explored aspect of the problem.
2. **Careful Mathematical Approach**: The authors present proofs and derivations systematically, with supporting lemmata and auxiliary proofs that strengthen the findings. This structured presentation is well-written, which helps readers follow the reasoning clearly through the mathematical arguments and supports the reliability of the results.

**Weakness:**

1. **Lack of Empirical Validation**: Although the paper’s theoretical contributions are well-developed, it lacks empirical validation or simulation studies to demonstrate the real-world applicability of the bounds. I understand that this conference primarily emphasizes theoretical perspectives; however, I believe that theory should ultimately enhance our understanding of practical models. Simulation studies could clarify how the derived bounds perform in practice, especially across diverse scenarios (e.g., naive Bayes and crowd-sourcing). Including a simulation section to demonstrate the tightness of the bounds across varying sensitivity and specificity conditions would allow readers to better assess the practical utility of these results.
2. **Limited Discussion of Practical Applications**: Although the theoretical results have potential applications in fields like crowd-sourcing and ensemble learning, there is minimal discussion of these possibilities. A more detailed application section would benefit readers interested in practical use cases. The authors briefly mention the relevance to decision theory but do not elaborate. Expanding on this could make the implications clearer for applied researchers.
3. **Dependence on Conditional Independence**: The assumption of conditional independence among experts is central to the derived bounds. It would be insightful to explore whether the bounds still hold when allowing for dependencies structured by a finite-dimensional covariate $Z$, i.e., $X_i\perp X_j\mid(Y,Z)$, where $Z$ denotes some finite-dimensional covariates.

**Minor issues**

1. **Complexity of Notation**: The paper makes extensive use of notation, which, while mathematically rigorous, could be redundant. For example, I think the introduction of $\gamma_i$ is not necessary in Theorem 3; explicitly write the bound with respect to $\pi_i/\bar{\pi}_i$ is easier to follow.
2. **Reference Style**: Ensure uniform citation styles throughout the document; some references seem to vary in formatting.

**Paper Award:**

No

---

### Official Review · Reviewer_p623 · 2024-11-08

**Rating:** 6
**Confidence:** 3

**Review:**

The paper is about proving (both upper and lower) error bounds for the optimal decision rule in the binary independent case when sensitivity and specificity are not the same. The authors provide three bounds, an expression for the error in terms of TV distance (Thm 1), and then an upper (Thm 2) and a lower (Thm 3) bound of the same expression.

The results look correct to me, and the paper is well written.

Let me also add that I am a bit surprised to hear that such results are original. I am not on top of this specific literature, but how do these results compare to standard lower bound approaches, e.g., Le Cam ?

**Paper Award:**

No

---

### Official Review · Reviewer_wMyh · 2024-11-09
**Review for Submisssion10**

**Rating:** 7
**Confidence:** 1

**Review:**

*Summary*

In this paper, the authors study the error probability of the optimal decision rule in the binary advice aggregation setting. This paper provides sharp upper and lower bounds in the asymmetric case, which sharpen the best known results in the symmetric special case.

*Strength*

The authors make concrete contributions to the binary advice aggregation setting by providing a novel and sharpened analysis of the error probability of the optimal decision rule.

*Weakness*

It would be better if the authors discuss more about the challenge of asymmetric setting compared to previously studied symmetric case.

**Paper Award:**

No

---

### Author Rebuttal · Authors · 2024-11-18

We thank the referees for the thoughtful and useful comments.

wMyh

>The authors make concrete contributions to the binary advice aggregation setting by providing a novel and sharpened analysis

Thank you for the kind words

>discuss more about the challenge of asymmetric setting compared to previously studied symmetric case

We'll be happy to elaborate on the analytical differences between the symmetric and asymmetric cases. In particular, we will explain in greater detail why the sharper lower bound for the symmetric case in Theorem 4 does not hold for the symmetric case, resulting in the weaker bound in Theorem 3

p623

>The results look correct to me, and the paper is well written.

Thank you for the kind words

>I am a bit surprised to hear that such results are original. I am not on top of this specific literature, but how do these results compare to standard lower bound approaches, e.g., Le Cam ?

We also found it surprising that such elementary results were not derived earlier. We do note that apparently the first bounds on the optimal decision error of *any* kind were derived by  Berend and Kontorovich in 2015 and sharpened by Manino et al. in 2019 -- so this whole line of inquiry appears to be relatively recent. Le Cam's methods apply in a more general setting of hypothesis testing, but do not yield sharper estimates in our setting.

7DKp
>Careful Mathematical Approach: The authors present proofs and derivations systematically, with supporting lemmata and auxiliary proofs that strengthen the findings. This structured presentation is well-written, which helps readers follow the reasoning clearly through the mathematical arguments and supports the reliability of the results.

Thank you for the kind words

>Lack of Empirical Validation
We understand the importance of empirical validation and will make an effort to include a simulation study for the camera version.


>Complexity of Notation

We will be happy to follow this suggestion


>The assumption of conditional independence

Indeed, conditional independence is a restrictive condition and our future research goals definitely include relaxing this assumption. We do not expect the bounds to hold verbatim when dependencies structured by a finite-dimensional covariate are allowed; some penality must be paid for the strength of these dependencies.

>Reference Style: Ensure uniform citation styles throughout the document

We will be sure to resolve this issue before the camera version.

---

> ### Comment · Reviewer_p623 · 2024-11-27
> **On Le Cam**
>
> Dear Authors,
>
> thanks for your responses. I should say that I am a bit puzzled by your answer about the relationship to Le Cam's method. You seem to be claiming that Le Cam is more general but it does not get better results than in this paper. What does this mean exactly ? Is Le Cam subsuming the results in this paper or not ? If not, can you please elaborate on it ?  [an elaboration on this point is what I was expecting in your rebuttal]
>
> Thanks

---

> > ### Author Response · Authors · 2024-11-27
> > **Le Cam's method**
> >
> > Thank you for the opportunity to clarify.
> >
> > There is no way to mathematically prove the claim, "technique X cannot yield result Y".
> >
> > Therefore, I cannot rigorously claim that Le Cam's method cannot possibly yield the novel bounds I derive in this paper.
> >
> > What I am quite certain about is these bounds have not previously appeared in print, and that they sharpen and generalize the best currently known bounds that have been published. I also note that every other approach to this problem in the literature did not proceed via Le Cam's method -- which seems to indicate that perhaps that is not the way to obtain optimal bounds in this case.

---

### Meta-Review · Area_Chair_erh6 · 2024-12-07

**Recommendation:** Accept
**Confidence:** 4

**Metareview:**

Meta-review

Context:
The authors study optimal tests of $Y=1$ or $Y=0$, given multiple independent tests $X_i$ ($1\leq i \leq n$) with $P(X_i=1|Y_i=1)=\psi_i$ and $P(X_i=0|Y_i=0)=\eta_i$. The optimal strategy is known since Parisi et al. (2014). Since then, many papers proved upper and lower bounds on the error of the error of the optimal test in the symmetric case $(\psi_1,...,\psi_n)=(\eta_1,...,\eta_n)$: Berend & Kontorovich (2015), Gao et al. (2016), refined by Manino et al. (2019). The authors focus here on the asymmetric case.

Contributions:
- upper bounds on the probability of error: the authors prove the 1st result in the asymmetric case: Thm 2, which recovers exactly Thm 1 of Manino et al. (2019) in the symmetric case.
- lower bounds: Thm 3 is the 1st bound in the general case. However, at least for large n, when applied to the symmetric case, it leads to results 1st than Thm 3 of Manino et al. (2019). So the authors also prove a new lower bound in the symmetric case, Thm 4, which improves the constants of Thm 3 of Manino et al. (2019).
The proofs of Thms 2-3-4 rely on an explicit formula for the probability of error given in Thm 1.

Thm 1 is to the general case what (5) is to the symmetric case. The authors claim that (5) is from Berend & Kontorovich (2015). Indeed, the proof of Thm 1 follows closely the proof of Lemma 2 in Berend & Kontorovich (2015), with a little extra care to deal with the asymmetry. This result is a useful tool to prove the inequalities in Thms 2-3-4, but it does not seem very relevant by itself. Maybe it would be better to state it as a lemma rather than as a Thm?
Thm 2 is not easy to prove. I insist that it is not obtained by extending the proof of Thm 1 of Manino et al. (2019). While the tools remain "elementary", the proof is tricky and original. The same comments apply to Thms 3 and 4. I checked the proofs carefully and they seem to be correct.

Clarity:
The paper is a little dry as the author makes little effort to engage the reader. The setting looks like a toy problem, while it can be motivated by its connections to Naive Bayes, or to practical problems such as crowdsourcing (this point is also raised by Reviewer 7DKp). On the other hand, the conciseness makes the paper very readable (Reviewer p623 shares my opinion on readability). The citations of existing works are accurate, the results are clearly stated and the proofs easy to follow.

Reviews:
The reviewers have positive opinions on the theoretical results, even though there is a huge variance in their confidence and in the depth of the arguments they provide to support the paper. They also raise a few questions:
- 7DKp regrets the lack of empirical validation. When it comes to tightening existing bounds, I share the opinion of the authors that this makes little sense. One could of course imagine numerical experiments to check if the final bounds are tight or if there is still room for improvement, but the authors already provide a clear identification of the regimes where this is the case (page 5).
- 7DKp regrets the lack of discussion on the potential applications. While a long discussion might impact the readability of the paper, in the "motivation" paragraph, I agree with him/her that it would not hurt to include a few keywords with references.
- wMyh asks what is so challenging in the asymmetric case. The authors proposed to add a discussion to explain why the sharper results of Thm 4 cannot be obtained in the asymmetric case, which would indeed be useful.
- Finally, p623 likes the theoretical results, but is also surprised that such results are novel and asks if they could be proven by Le Cam's method.

Le Cam's method allows to prove lower bounds on the risk of statistical estimators, in two steps: 1) reduction to a test problem 2) lower bound on the test error. For step 2) I will refer to a classical reference on the topic: [Tsybakov, Introduction to Nonparametric Estimation]. There step 2) is tackled by Thm 2.2 (i) (due to Le Cam, 1973), where the test error is lower bounded in terms of the total variation distance (TV) between the hypotheses. Then, one must still control the total variation itself: this is the part left to the user, as it depends on the model studied. Let's call it step 3).

In this paper, the authors don't have to use step 2) above, because Thm 1 directly characterizes the error of the optimal test in terms of TV. Moreover, Thm 1 is an equality, not an inequality as in the general case, due to the specific context they consider. What the authors do here is step 3): they study accurately the TV norm itself. Even if we use Le Cam's method instead of Thm 1, this would still be a necessary step.

Conclusion:
Overall, this is an elegant improvement of existing theoretical results.

Minor comment:
- page 1: it is written that $X_i$'s are iid but their distribution is actually not the same ($\psi_i$ depends on $i$!), they are only independent.

**Paper Award:**

No